# Testosterone in Males as Enhanced by Onion (*Allium Cepa* L.)

**DOI:** 10.3390/biom9020075

**Published:** 2019-02-21

**Authors:** Saleem Ali Banihani

**Affiliations:** Department of Medical Laboratory Sciences, Jordan University of Science and Technology, Irbid 22110, Jordan; sabanihani@just.edu.jo; Tel.: +962-2-720-1000

**Keywords:** onion, testosterone, luteinizing hormone, oxidative stress, antioxidants

## Abstract

Testosterone (17β-Hydroxyandrost-4-en-3-one) is the main sex hormone in males. Maintaining and enhancing testosterone level in men is an incessant target for many researchers. Examples of such research approaches is to utilize specific types of food or dietary supplements as a safe and easily reached means. Here, specifically, since 1967 until now, many research studies have revealed the effect of onion on testosterone; however, this link has yet to be collectively reviewed or summarized. To accomplish this contribution, we searched the Scopus, Web of Science, and PubMed databases for full articles or abstracts (published in English language) from April 1967 through December 2018 using the keywords “onion” versus “testosterone”. In addition, a number of related published articles from the same databases were included to improve the integrity of the discussion, and hence the edge of the future directions. In summary, there is an evidence that onions enhance testosterone level in males. The mechanisms by which this occurs is mainly by increasing the production of luteinizing hormone, enhancing the antioxidant defense mechanism in the tests, neutralizing the damaging effects of the generated free radicals, ameliorating insulin resistance, promoting nitric oxide production, and altering the activity of adenosine 5′-monophosphate -activated protein kinase. However, this effect requires further approval in humans, mainly by conducting clinical trials.

## 1. Introduction

In general, testosterone (17β-Hydroxyandrost-4-en-3-one) is the main sex hormone in humans and other species males [1]. It plays a key role in developing their reproductive organs and other sexual features such as body hair and muscle mass [2]. Decreased levels of testosterone in males are linked with a wide range of diseases/disorders such as diabetes mellitus [3], male infertility [4,5,6], Alzheimer’s [7], osteoporosis [8], depression [9], and cardiovascular disease [10,11]. Therefore, various research studies as well as reports have focused on preserving and enhancing testosterone in males. Examples of this specific research intention included the use of certain types of food or dietary supplements.

The onion (*Allium cepa* L.), also known as common onion or bulb onion, is a vegetable that has been grown and selectively farmed for more than seven thousand years [12]. At this moment in time, onions become part of humans’ food; for example, in 2016, the world production of onions (dried onions) reached more than 90 million tons, led by China (~26% of the total) and India (~21% of the total). 

Naturally, onions can be found in three main subtypes: white, red, and yellow. Regardless of the subtype, in general, raw onion bulbs contain approximately 89% water, 4% sugar, 2% dietary fiber, 1% protein, a negligible amount of fat (~100 mg/100 g), and low amounts of essential nutrients [13]. Accordingly, the energy value of onion is not significant (~40 kcal/100 g) [13]. Most importantly, onions were found to contain several and unique phytochemicals such as quercetin and quercetin glucosides, kaempferol, thiosulfinates, cepaenes, anthocyanins, and sulfur compounds (e.g., S-methyl cysteine sulfoxide, diallyl trisulfide, S-allyl cysteine sulfoxide, dimethyl trisulfide) that have been identified to have an impact on humans’ health [14,15]. Yet, some of these phytochemicals are under basic and translational research to explore their possible biological effects in humans.

It is worth mentioning that Okinawans, the longest-living people out of any state or country in the world according to the World Health Organization reports, were found to consume habitually considerable amounts of certain vegetables including onions [16]. Besides, surprisingly, serum testosterone was found to be higher in Okinawan men compared, for examples, with the age-matched Americans [16]. This could be an evidence that onions may have a potential impact on general health in men.

As per Scopus database, since 1967 until now, there are at least 27 research articles linked onion to testosterone; however, this link has not yet been collectively reviewed and summarized. Here, we searched the Scopus, Web of Science, and PubMed databases for full articles or abstracts (published in the English language) from April 1967 through December 2018 using the keywords “onion” and “testosterone”. In addition, selected indirect published articles from the same databases were included to reach the comprehensive and integral discussion and conclusion. 

## 2. Effect of Onions on Testosterone 

Table 1 summarizes the main and direct studies conducted on onions or onion extracts and their reported effects on testosterone level. Almost all of these studies were conducted in vivo utilizing rat models, particularly male rats. Only one study was conducted on healthy men. 

In addition, as shown in Table 1, the effect of onions on testosterone level was often measured using onion juice or aqueous onion extracts rather than directly using the onion bulb; only one study has utilized onion bulb. In the animal studies, considering the studies conducted on onion juice, the minimum used dose was approximately 0.5 mL rat^−1^ day^−1^ and the maximum used dose was approximately 6 mL rat^−1^ day^−1^. Onion juice, extract, or bulb were given orally to all tested in vivo systems over a period ranging from 10 to 60 days. 

As a result, in general, the main stream of research (75% of research studies) that are directly linking between onion and testosterone reveal a positive effect of onion on testosterone level in males. The rest of these studies (25%) did not show a significant effect of onion on testosterone level. These blunted studies were mainly conducted on male rats induced by chemical toxicants (cadmium, aluminum), which may induce irreversible damage to the reproductive organs. 

Specifically, as shown in Table 1, it is important to point that the only human study that directly link between onion and testosterone reveal a positive effect of onion extracts on testosterone [17]. However, more confirmatory human studies in this specific research context are still very imperative.

The summarized evidence from this study supports the use of onion as an aphrodisiac food, to increase libido and strengthen the reproductive organs, by many cohorts [18]. Testosterone therapy was used to treat hypoactive sexual desire [19]. Traditionally, fried onions in a pure butter mixture was used by many cohorts, particularly Indians, as an aphrodisiac tonic, especially when taken with a spoon of honey [18]. In addition, another Indian traditional mixture called kanji, which is a powder of black gram dipped in onion juice for seven days and then dried, was used as an aphrodisiac food [18].

## 3. Mechanistic Studies

### 3.1. Effect of Onion on Luteinizing Hormone

In men, testosterone is mainly synthesized in Leydig cells [32]. The function and the number of Leydig cells in the testis in turn is regulated primarily by luteinizing hormone and secondarily by follicle-stimulating hormone [32]. The amount of the produced testosterone by Leydig cells is under the control of luteinizing hormone [33,34]. Specifically, luteinizing hormone regulates the expression of 17β-hydroxysteroid dehydrogenase, which catalyzes the conversion of androstenedione to testosterone. The formed testosterone is transported to Sertoli cells in the testis to enhance sperm production (i.e., spermatogenesis) [35].

A number of studies have revealed that onions have a positive impact on production of luteinizing hormone; for example, Wistar male rats administered fresh onion juice at 1 g rat^−1^ day^−1^, for 20 consecutive days, had higher levels of luteinizing hormone compared with the control [21]. In addition, aqueous onion extract (1 mL of the extract/100 g of body weight, for eight weeks) significantly increased luteinizing hormone in both normal and aluminum chloride-treated male rats [25]. Accordingly, the positive effect of onion on testosterone may be, at least in part, attributable to the increased level of luteninizing hormone. 

### 3.2. Effect of Onion as a Potential Antioxidant on Testosterone

Generally, in cellular systems, accumulation of reactive oxygen species and formation of oxidative stress state has been found to significantly reduce the function of the cell [33,36]. Such reduction may be due to the increased oxidative damage to the functional and structural components (e.g., proteins, lipids, and nucleic acids) of the cell. Therefore, countering this oxidative damage should convalesce the performance of the cell [37]. 

In particular, increased generation of reactive oxygen species, and hence the oxidative damage in testicular Leydig cells may reduce the synthesis of testosterone [10]. This reduction may be aggregated in the presence of chemical toxicants such as aluminum and cadmium, and in the presence of certain drugs such as permethrin, which is a medication and insecticide used to cure scabies. Indeed, aluminum chloride was found to significantly decrease testosterone synthesis [25], may be by reducing the activity of the testicular antioxidant enzymes such as superoxide dismutase, catalase, and glutathione reductase, and disrupting the gene expression of 3β-hydroxysteroid dehydrogenase and cholesterol side-chain cleavage enzyme in the testis [38]. It has been shown that the enzymatic antioxidants were significantly increased in onion treated-male rats and in onion treated-male rats with aluminum-induced reproductive toxicity [25]. Such enhancement for the antioxidant defense mechanism in Leydig cells by the effect of onion may increase the production of testosterone. In addition, onion was found to decrease the formation of malondialdehyde, a marker of oxidative damage, particularly lipid peroxidation, in normal male rats and male rats with aluminum-induced reproductive toxicity [39]. Moreover, aqueous onion extract was found to protect against cadmium-induced oxidative stress; however, it did not recover cadmium-induced testosterone depletion [27].

Furthermore, in rats, permethrin was found to significantly reduce the number of Leydig cells and the amount of testosterone produced [30]. Via unknown pathway, onion juice (3 mL rat^−1^, for 60 days) was able to decrease damage to the Leydig cells and improve testosterone production [30]. The mechanism by which this occurs is mainly by reducing the oxidative damage to Leydig cells; given that onion contains potent antioxidant molecules such as quercetin and quercetin glucosides, thiosulfinates, and anthocyanins [30]. 

Actually, among 28 vegetables, onions were found to rank the highest in quercetin content. In effect, quercetin and quercetin derivatives have been found to have an impact on human’s health. It is important mention that quercetin has been found to protect against wide-range of diseases/disorders such as asthma [40], diabetes [41], gout [42], cancer [43], arthritis [44], osteoporosis [45], and neurodegenerative disorders [46]. Therefore, the high content of the antioxidant quercetin and its derivatives in onion makes it very effective dietary contrivance to enhance humans’ health. 

Specifically, quercetin (50 mg kg^−1^ day^−1^, for 15 days) significantly increased testosterone level in arsenic-induced reproductive damage in rats [47]. In addition, quercetin at 90 mg^−1^ kg^−1^ day^−1^ for 15 days was found to enhance testosterone production in male rats with di-(2-ethylhexyl) phthalate-induced reproductive toxicity [48]. Moreover, adult male albino rats with atrazine-induced reproductive toxicity administered quercetin at 10 mg^−1^ kg^−1^ of body weight for 21 days had higher levels of testosterone compared with the control [49]. Furthermore, quercetin at 20 mg^−1^ kg^−1^ of body weight for two weeks restored plasma testosterone induced by sulphasalazine in male rats [50]. Recently, it has been shown that quercetin at 20 mg^−1^ kg^−1^ of body weight for 28 days is able to attenuate testosterone depletion induced by cadmium chloride in male Wistar rats [51]. Such protective effects may be attributable to the antioxidant activity of quercetin as it was found to increase the activity of testicular antioxidants (superoxide dismutase, glutathione reductase, and catalase) and significantly reduce the level of malondialdehyde in testicular tissue [47,50,52].

Moreover, it was concluded that vegetables, rich in quercetin-4′-glucoside, such as onion, are likely to serve as valuable antioxidant sources for reducing iron-induced oxidative stress and lipid peroxidation in live cells [53]. In addition, in general, the aqueous extract of onion has been identified as having a potent hydroxyl radical scavenging activity utilizing 2,2-Diphenyl-1-picrylhydrazyl [54]. Furthermore, *A. cepa* L. polysaccharide fractions were found to have a strong antioxidant actions toward 3-ethylbenzothiazoline-6-sulphonic acid radical cations, superoxide anion radical scavenging, and Fe^2+^ chelating [55]. 

Further, the reduction in glutathione may diminish the detoxification process of the produced oxidants in the testes, which delays or reduce the synthesis of testosterone [56]. It was shown that quercetin prevented testicular glutathione depletion, thereby resulting in a decrease in content prooxidants [52]. Such protective effects may contribute to enhance the production of testosterone, and hence the ways of its action. 

### 3.3. Effect of Onion as an Antihyperglycemic Affecter on Testosterone

Several studies have revealed that the generation of free radicals, and hence the level of oxidative damage, in diabetic conditions, most of time, is higher than in normal conditions [57,58,59]. Such hyperglycemic-induced oxidative stress states was found to delay or disrupt the function of the cell [57,60]. 

It was demonstrated that that onion has hypoglycemic effects and can be used as a dietary supplement to manage the progression in patients with type 1 and type 2 diabetes [61]. In addition, onion peel extracts, rich with quercetin, was found to reduce insulin resistance in male diabetic rats on high fat diet [62]. Moreover, streptozotocin-induced diabetic rats administered onion and fenugreek had lower oxidative stress levels compared to those administered only fenugreek [63]. In addition, onion juice was found to have an insulin-like action as it decreased blood glucose level in hyperglycemic animal models [20]. Moreover, one of the major bioactive compounds in onion are sulfur compounds such as S-methyl cysteine sulfoxide, diallyl trisulfide, S-allyl cysteine sulfoxide, dimethyl trisulfide, dipropyl disulfide, propenyl propyl disulfide, propenyl methyl disulfide, and methyl propyl trisulfide [15]. It has been shown that many of these bioactive compounds have insulinotropic as well as antidiabetic activity in animal models, which may explain an improvement in beta-cell function and increase in insulin sensitivity [15]. Actually, these sulfur compounds in onions were found to enhance the acticity of hepatic enzymes (e.g., glucose-6-phosphatase and glucokinase), which could be, at least in part, behind the antihyperglycemic effect of onion in the experimental animals.

Collectively, according to these evidences, it can be suggested that the hypoglycemic effect of onion has a positive effect on testosterone production, particularly in diabetic conditions. Indeed, streptozotocin-induced diabetic rats orally administered onion at 15 mg kg^−1^ day^−1^ for four weeks had higher testosterone levels compared with the control [22]. 

### 3.4. Effect of Onion as a Nitric Oxide Stimulator on Testosterone

It is well-known that the decrease in blood flow to the testis reduces the synthesis of testosterone, which negatively affect spermatogenesis [64,65]. Enhancing nitric oxide, a free radical gas generated by nitric oxide synthase and acts as a vasodilator, may increase blood flow in the testis and promote testosterone synthesis [33]. It has been suggested that aqueous onion extract has nitric oxide-releasing properties [66]. Therefore, enhancing nitric oxide production could be a contributing factor behind the reported positive effects of onions on testosterone level. 

### 3.5. Effect of Onion on 5′ AMP-Activated Protein Kinase

5′ AMP-activated protein kinase is an enzyme (heterotrimeric protein-complex) plays a vital role in energy homeostasis in cellular systems; specifically, by enhancing fatty acids oxidation and glucose uptake [67]. It has been shown that 5′ AMP-activated protein kinase activates the production of testosterone in Leydig cells in the testes [68]. In general, studies have shown that, quercetin, a functional molecule in onion, activates 5′ AMP-activated protein kinase [69,70]. Therefore, it can be suggested that onion may induce testosterone production by enhancing 5′ AMP-activated protein kinase. However, this suggestion requires approval by further research studies. 

## 4. Conclusions and Future Perspectives

Collectively, there is an evidence that onion or onion extracts (e.g., aqueous extract ~30 mg day^−1^) enhances testosterone production in males. However, this effect requires further approval in humans, mainly by conducting clinical trials. 

The mechanisms by which onion enhances testosterone production in males is mainly by enhancing the production of luteinizing hormone, neutralizing the damaging effects of the formed free radicals, mainly in the testes, enhancing the antioxidant defense mechanism (e.g., antioxidant enzymes, glutathione) in the testis, ameliorating insulin resistance, promoting nitric oxide production in Leydig cells, and altering the activity of 5′ AMP-activated protein kinase (Figure 1).

## Figures and Tables

**Figure 1 biomolecules-09-00075-f001:**
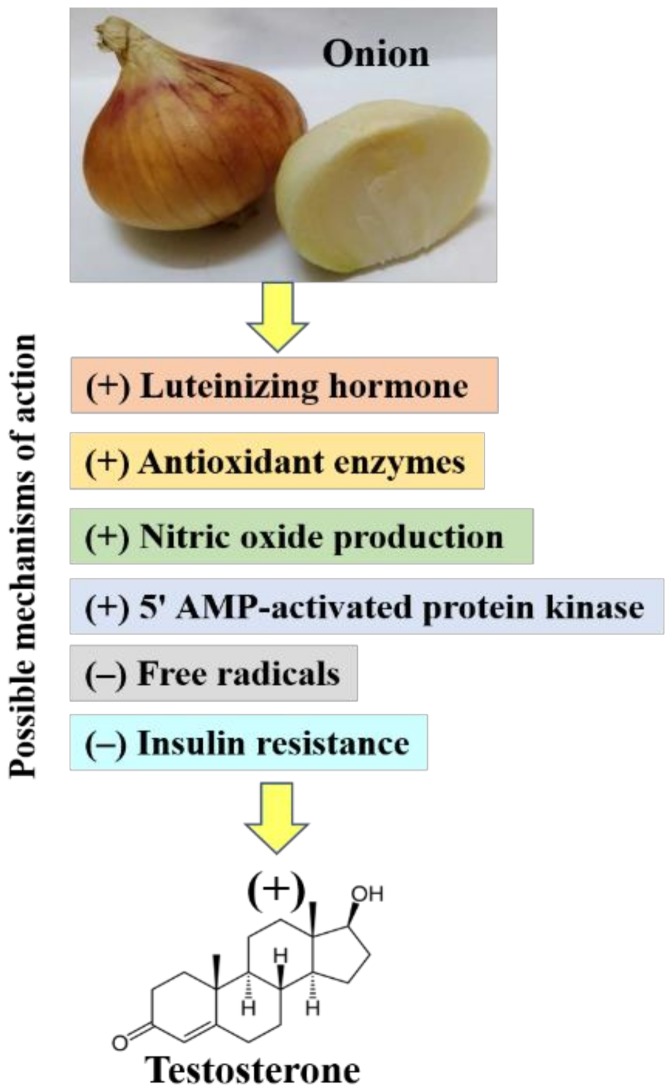
Possible mechanisms by which onion enhances testosterone production in males.

**Table 1 biomolecules-09-00075-t001:** A summary of the research studies conducted on onions or onion extracts and their reported effects on testosterone.

(Onion or Onion Derivative)	Dose (Mode of Treatment)	Duration	Population	Effect on Testosterone	Reference
Onion juice	2, 4, and 6 mL day^−1^ (Orally)	10 days	Male albino rats	(+)	[20]
Onion juice	0.5 and 1 g rat^−1^ day^−1^ (Orally)	20 days	Wistar male rats	(+)	[21]
Onion	15 mg kg^−1^ day^−1^ (Orally)	4 weeks	Streptozotocin-induced diabetic	(+)	[22]
Onion juice	3 mL day^−1^ rat^−1^ (Orally)	4 weeks	Lamotrigine-induced male rats	(+)	[23,24]
Aqueous extract of onion	1 mL of the extract/100 g of body weight (Orally)	8 weeks	Male rats with aluminum-induced reproductive toxicity	(±)	[25]
Aqueous extract of onion	1 and 2 mL rat^−1^ (Orally)	4 weeks	Rats	(±)	[26]
Aqueous extract of onion	1 mL/100g of body weight (Orally)	4 weeks	Male rats with cadmium-induced organ toxicity	(±)	[27]
Onion juice	1 and 2 mL day^−1^ rat^−1^ (Orally)	20 days	Adult male abino rats	(+)	[28]
Onion juice	3 mL rat^−1^ (Orally)	60 days	*Eschericcia coli* infected male rats	(+)	[29]
Onion juice	3 mL rat^−1^ (Orally)	60 days	Permethrin-induced male rats	(+)	[30]
Onion juice	40 mg kg^−1^ (Orally)	14 days	Testicular torsion/detorsion rat models	(+)	[31]
Onion extracts containing concentrated cysteine sulfoxides	30 mg day^−1^ (Orally)	14 days	Healthy men	(+)	[17]

(+) Increase, (−) Decrease, (±) no effect.

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
