# Peer review of "Testosterone in Males as Enhanced by Onion (Allium Cepa L.)"

_biomolecules, 2019, doi:10.3390/biom9020075_

Round 1
Reviewer 1 Report
Please see attached file

Author Response
Comment 1-Line 34
1- Sönmez, M., Türk, G. and Yüce, A., 2005. The effect of ascorbic acid supplementation on sperm quality, lipid peroxidation and testosterone levels of male Wistar rats. Theriogenology, 63(7), pp.2063-2072.
2- Dohle GR, Smit M, Weber RF. Androgens and male fertility. World journal of urology. 2003 Nov 1;21(5):341-5.
3- Su LM, Goldstein M, Schlegel PN. The effect of varicocelectomy on serum testosterone levels in infertile men with varicoceles. The Journal of urology. 1995 Nov 1;154(5):1752-5.
4- Du Plessis SS, Cabler S, McAlister DA, Sabanegh E, Agarwal A. The effect of obesity on sperm disorders and male infertility. Nature Reviews Urology. 2010 Mar;7(3):153.
Response: Thank you very much for these suggested references. We have added 2 of these references.
Comment 2 – line 40.
Brewster JL. Onions and other vegetable alliums. CABI; 2008.
Response: Thank you very much for this comment. We have added this reference and removed our reference.
Comments 3,4,5- lines 62, 64, 66
Reference??
Response: Thank you very much for this comment. Actually, this is results and has to be extracted from table 1 and its related references. We have added “as shown in table 1” and references when required.
Comment 6 – line 75
Is not good enough. Not well written
Response: Thank you very much for this comment. We have reworded these sentences and integrated the reference.
Comment 7
is not complete
1- Khaki A, Farnam A, Badie AD, Nikniaz H. Treatment effects of onion (Allium cepa) and ginger (Zingiber officinale) on sexual behavior of rat after inducing an antiepileptic drug (lamotrigine). Balkan medical journal. 2012 Sep;29(3):236.
Thank you very much for this comment. Actually, we were cacious about this study as it is very similar to the one that we already cited by the same authors, but published in another journal. However, we have cited this suggested paper.
Khaki, A., Farnam, A., Ahmadi-Ashtiani, H.R., Rezazadeh, S., Rastgar, H., Eftekharzadeh, S., Aghamohamadi, R., Abiri, N. . Treatment effect of onion on sexual behavior after induces an antiepileptic drug (lamotrigine) in male rat. Journal of Medicinal Plants 2010, 9, 49-57.
Comment 8-Line 100
Not well written.
Response: Thank you very much for this comment. We have fixed this sentence.
Comment 9 – Line 139.
Izawa H, Kohara M, Aizawa K, Suganuma H, Inakuma T, Watanabe G, Taya K, Sagai M. Alleviative effects of quercetin and onion on male reproductive toxicity induced by diesel exhaust particles. Bioscience, biotechnology, and biochemistry. 2008 May 23;72(5):1235-41.
Response: Thank you very much for this suggestion. While, this article does show the effect of onion on testosterone, it shows the effect on sperm function. Therefore, we feel it is not suitable in this location as we specifically express the effect of onion on testosterone.
Comment 10 – Line 172.
Ige SF, Akhigbe RE. The role of Allium cepa on aluminum-induced reproductive dysfunction in experimental male rat models. Journal of human reproductive sciences. 2012 May;5(2):200.
Response: Thank you for this comment. We have already integrated this reference.
Comment 12-Line 187
Response: Thank you very much for this comment. Actually, we modified and moved the whole paragraph to the introduction as requested by other reviewers. Again, thank you for your comment.
Comment 13, 14, 15, 16, 17, 18, 19 in the references.
Response: Thank you very much for your comments. We have fixed all incorrected styles.
Reviewer 2 Report
Dear
I think this article could accept after minor English edition .
Regards
Author Response
I think this article could accept after minor English edition.
Response: Thank you very much for your comments. The article was carefully reviewed by a native English scientist (changes were marked blue).
Reviewer 3 Report
Considering the high impact factor of the journal, the article need a deep revision and restored. And better presentation of paragraphs.
In general, when cited testosterone should be included "level" Line 58, 68 etc.
Line 48. describe the active phytochemical contained in onions. Quercitine ?
line 72. Which is the study? It is meaningless expression.
Line 76-77 In the Kumar K.P.S. 2010 (13) is not well interpeted. The study decribe the effect of quercetin protects against cataracts, cardiovascular disease, and cancer. and other organosulfur compounds with anti-cholesterol effect are described. Furthermore, considering the anti-inflammatory and hypoglycaemic effect onion buld, could the increased level of plasma testosterone be a consequence of metabolic changes?
Table 1. It is necessary to report the changes of testosterone level and the significativity.
in the second paragraph (line 57) should be included the mechanistic studies explaining and completing the mechanism of action of onion at molecular and cellular level. Also the active form of polyphenols that are considered active in onion.The text from line 91 to 97 is not necessary.
Line 98: A number of studies.....referencer are missing and incomplete.
Paragraph 3.2 needs to be reorganized focusing the antioxidant effects of onion. Line 106-110 The concept is generic and should be explained in reference to testosterone production. Reference 28 in this case is not appropriate.
Line 111 the synthesis of testosterone is not inhibited only from reactive oxygen species, but other metabolic factors should be mentionated.
Line 118: studieshave shown that....Only one study is cited and should be properly mentionated.
paragraph 3.3 It should be rewritten. In Diabetic condition not only oxidative stress is resposible of tissue damage, but also hormonal factors related to nutrition (insulin, IGF1, testosterone, etc)
line 160: here you reach a conclusion, but explanation and references are missed. Which are the bioactive molecules that have hypoglycemic effect? Are they independenf of the diet?
line 162: It was evident ...to be romoved. It was demonstrated that...
paragraph 3.4. this is another potential mechanism of action (enhanig the nitric oxyde-releasing) of onion that is considered for cardiovascular and atherosclerosis protection, but if you think is involved in testosterone production, please put the specific references.
in paragraph 5. the Okinawa study should be better considered and cited in the introduction. This type of diet evaluated the effect of low low caories diet, low glycemic load and the level of testosterone can be correlated mor with the diet or onion intake?
In the conclusions should be given the essential informations about the suspected activity of onion bulb on testosterone plasma level, the dose of assumption suggested, the critical points about the prevalence of studies conducted in animal, and finally if there is a diet to associate with onion.
Author Response
Considering the high impact factor of the journal, the article needs a deep revision and restored. And better presentation of paragraphs.
Response: Thank you very much for this comment. We have made a further detailed revision for the article and made professional improvements in the whole article, including: addition of original conclusive figure, adding more specific details to almost all sections of the article, upgrade references (we have discussed and added 8 more references), and enhancing the English by a native English scientist. (additions/modifications were marked blue). In addition, we have responded to 4 potential reviewers as they requested several modifications.
In general, when cited testosterone should be included "level" Line 58, 68 etc.
Response: Thank you very much for this comment. We have integrated the term “level” as requested.
Line 48. describe the active phytochemical contained in onions. Quercitine?
Response: Thank you very much for this comment. We have provided further information on the main bioactive molecules in Allium cepa and discussed more references. (third paragraph in the introduction-marked blue).
line 72. Which is the study? It is meaningless expression.
Response: Thank you very much for this comment. Actually, this is the only human study that is presented in table 1, we have added the required information and defined this study.
Line 76-77 In the Kumar K.P.S. 2010 (13) is not well interpeted. The study decribe the effect of quercetin protects against cataracts, cardiovascular disease, and cancer. and other organosulfur compounds with anti-cholesterol effect are described.
Response: Thank you very much for this comment. Actually, we are aware that this reference provides several health benefits for onion, while we were specifically focusing in this work to discuss only the effect of onion on testosterone and its related biological function. Therefore, we only extracted the related information from this reference which serves our aim and the topic of this section which is “Effect of onions on testosterone”. Again, thank you very much for your deep discussion and review.
Furthermore, considering the anti-inflammatory and hypoglycaemic effect onion buld, could the increased level of plasma testosterone be a consequence of metabolic changes?
Response: Thank you very much for this comment. Yes, it can be a consequence of metabolic change as there is equilibrium between testosterone and general health, in which general health enhances testosterone and vice versa testosterone enhances general health. Actually, we have already discussed this, but in brief, in the second paragraph (section 3.3), and as of the comment by our reviewer, we have reorganized this section and added more discussion.
Table 1. It is necessary to report the changes of testosterone level and the significativity.
Response: Thank you very much for this comment. In general, as it is well-known in research studies, the significative change in testosterone level means that there is a statistical significance (mainly: P value is less than 0.05 between the test and the control). However, if our potential reviewer requested that we should extract this P value from each study and add it in the table, we should do so.
in the second paragraph (line 57) should be included the mechanistic studies explaining and completing the mechanism of action of onion at molecular and cellular level. Also, the active form of polyphenols that are considered active in onion.
Response: Thank you very much for this comment. Actually, in this section we are strictly and systematically explained the main direct studies that are linking between testosterone and onion as per our searching methodology (summarized in table 1), and we did not skip answering our potential reviewer question, since the mechanisms of action of onion and its main bioactive compounds at molecular and cellular levels are explained in details in the mechanistic studies sections 3.1, 3.2, 3.3, 3.4, 3.5. And as per our potential reviewer comments, we have enhanced these sections with further details considering the main molecular mechanisms that may explain enhancing testosterone production by the effect of onion and its bioactive compounds. (all additions were marked blue). Again, thank you for this comment as we have discussed and added more references.
The text from line 91 to 97 is not necessary.
Response: Thank you for this comment. We added this text to provide the readers with information that testosterone is controlled by another hormones, mainly LH, and we felt that it is important to provide a basic knowledge (source of production and function) about LH hormone in order to integrate the effect of onion in this specific biological context. However, if this is mandatory, we will delete this text.
Line 98: A number of studies.....referencer are missing and incomplete.
Response: Thank you very much for this comment. Actually, these references are explained in the following text as we have typed in the second sentence “for example.”; however, we merged the first 2 sentences together, and already have started with “in addition” in the second sentence.
Paragraph 3.2 needs to be reorganized focusing on the antioxidant effects of onion.
Response: Thank you very much for this comment. Actually, in this paragraph we intended to particularly discuss where onion as an antioxidant enhances testosterone production using the direct studies in this specific context; therefore, we have provided a background (only in the first paragraph and in the beginning of second paragraph) for the reader and then pointed for the studies that are directly interplay in this context. However, as of the intention by our potential reviewer we have added more specification on the antioxidant activity of onion and added 3 more references (changes were marked blue).
Line 106-110 The concept is generic and should be explained in reference to testosterone production. Reference 28 in this case is not appropriate.
Response: Thank you very much for this comment. We have enhanced the references in this concept and added another reference.
Line 111 the synthesis of testosterone is not inhibited only from reactive oxygen species, but other metabolic factors should be mentioned.
Response: Thank you very much for this comment. Yes, actually we are aware that other factors rather than reactive oxygen species may affect testosterone synthesis. While, we intended here to explain the oxidative stress mechanism as this is the main mechanism that has been specifically and experimentally studied by the specific studies that are directly linking between onion and testosterone according to our methodology in this study and searching key words (testosterone, onion). However, if this is mandatory, we will do so and write on other specific mechanism, but this may not be directed to the effect of onion on testosterone and we feel it may confuse the reader. Again, thank you very much for this comment.
Line 118: studies have shown that....Only one study is cited and should be properly mentionated.
Response: Thank you very much for this comment. We have reworded this sentence as it is specific and it won’t be suitable to increase the number of the references. Again, thank you very much for this comment.
paragraph 3.3 It should be rewritten. In Diabetic condition not only oxidative stress is resposible of tissue damage, but also hormonal factors related to nutrition (insulin, IGF1, testosterone, etc).
Response: Thank you very much for this comment. Yes, it is not only oxidative stress, we have added further explanation and discussed more references. Again, thank you very much for this comment.
line 160: here you reach a conclusion, but explanation and references are missed. Which are the bioactive molecules that have hypoglycemic effect? Are they independenf of the diet?
Response: Thank you very much for this comment. Actually, in this conclusive sentence we were preparing the reader to the idea that certain types of food may have impact on hyperglycemic conditions, and one such example is onion. While, yes, we have reviewed this conclusive sentence and found to delete it as it is not suitable for our specific idea.
line 162: It was evident ...to be removed. It was demonstrated that...
Response: Thank you very much for this comment. We have made this change.
paragraph 3.4. this is another potential mechanism of action (enhanig the nitric oxyde-releasing) of onion that is considered for cardiovascular and atherosclerosis protection, but if you think is involved in testosterone production, please put the specific references.
Response: Thank you for this reference. Actually, there is no direct reference, and we logically suggest this effect by testosterone based on other affecters such as onion. However, we added this indirect reference to the second sentence.
in paragraph 5. the Okinawa study should be better considered and cited in the introduction. This type of diet evaluated the effect of low low caories diet, low glycemic load and the level of testosterone can be correlated mor with the diet or onion intake?
Response: Thank you very much for this comment. We have moved this study to the introduction. Actually, onion is one of the main vegetables that are habitually consumed by the Okinawans.
In the conclusions should be given the essential information about the suspected activity of onion bulb on testosterone plasma level, the dose of assumption suggested, the critical points about the prevalence of studies conducted in animal, and finally if there is a diet to associate with onion.
Response: Thank you very much for this comment. We have reorganized the conclusion as suggested by other reviewers and we draw a figure to summarize this conclusion. In addition, we added some of the requested information as suggested, while we could not fully add all as there is only one human study on the effect of onion on testosterone and, when taken alone, it may provide a week conclusive information. Actually, in our conclusion we have pointed for the lack of human studies in this research context, which is a gap of information and could be a hot nutritional research approach.
Reviewer 4 Report
Overall the paper presents some interesting information the review topic is original and worthy of investigation. The paper is well written and structured however I recommend making the following changes to improve the current version of the paper:
Title: I think it will be better to replace “affected” by “enhanced” or “improved”. since affected is related to negative effect
Line 10-15: in the Abstract, several introductive sentences should be moved to the introduction section
Line 39: Please use italic for Latin names Allium cepa L.
Line 48-50: the section related to phytochemicals from Allium cepa is very limited. Authors should provide further information on the main bioactive molecules from this specie.
Page 4: “3. Mechanistic studies” The mechanistic studies is detailed in the text however to attract readers and increase citations of this review paper it is necessary to draw a graph or table with the main mechanism of action (the cascade of actions) of Allium cepa extracts (mains compounds) with direct or indirect effect on male testosterone.
Author Response
Overall the paper presents some interesting information the review topic is original and worthy of investigation. The paper is well written and structured however I recommend making the following changes to improve the current version of the paper:
Response: Thank you very much.
Title: I think it will be better to replace “affected” by “enhanced” or “improved”. since affected is related to negative effect
Response: Thank you very much for this comment we have made this required change.
Line 10-15: in the Abstract, several introductive sentences should be moved to the introduction section.
Response: Thank you very much for this comment. We have deleted most of the introductory sentences from the abstract. While we did not move this deletion to the introduction as similar sentences or ideas are already present there, and this will be redundancy. Again, thank you very much.
Line 39: Please use italic for Latin names Allium cepa L.
Response: Thank you very much. We have made this change.
Line 48-50: the section related to phytochemicals from Allium cepa is very limited. Authors should provide further information on the main bioactive molecules from this specie.
Response: Thank you very much for this comment. We have provided further information on the main bioactive molecules in Allium cepa. (third paragraph of the introduction-marked blue).
Page 4: “3. Mechanistic studies” The mechanistic studies is detailed in the text however to attract readers and increase citations of this review paper it is necessary to draw a graph or table with the main mechanism of action (the cascade of actions) of Allium cepa extracts (mains compounds) with direct or indirect effect on male testosterone.
Response: Thank you very much for this comment. We have integrated this requested graph in the conclusion section.
Round 2
Reviewer 1 Report
Dear Authors
I believe that the manuscript in its present form is suitable for publication in Biomolecules.
Regards
Reviewer 3 Report
The title: Testosterone in male as or "Testosterone level in male is affected
in the abstract: The mechanisms by which this occurs is mainly by increasing
the word "due" is missing:
The mechanisms by which this occurs is mainly due by increasing
Please check the style in the text.